# Ultra-Short Peptide Hydrogels as 3D Bioprinting Materials

**DOI:** 10.3390/gels12010049

**Published:** 2026-01-02

**Authors:** Davina In, Androulla N. Miliotou, Panoraia I. Siafaka, Yiannis Sarigiannis

**Affiliations:** 1College of Humanities and Sciences, Virginia Commonwealth University, Richmond, VA 23284, USA; davina.thidain@gmail.com; 2Department of Health Sciences, School of Life & Health Sciences, University of Nicosia, Nicosia 2417, Cyprus; miliotou.a@unic.ac.cy; 3Bioactive Molecules Research Center, School of Life & Health Sciences, University of Nicosia, Nicosia 2417, Cyprus; 4Department of Life Sciences, School of Sciences, Program of Pharmacy, European University Cyprus, Nicosia 2404, Cyprus; p.siafaka@euc.ac.cy

**Keywords:** ultra-short peptides, hydrogels, bioprinting, bioink, biomaterials

## Abstract

Ultra-short peptides (USPs; ≤7–8 amino acids) emerge as minimal self-assembling building blocks for hydrogel-based biomaterials. Their intrinsic biocompatibility, straightforward synthesis, and ease of tunability make them particularly attractive candidates for potential use in bioprinting. This review provides an overview of the properties of USPs along with their applications in three-dimensional (3D) bioprinting. We first discuss how peptide sequence, terminal and side-chain modifications, and environmental triggers govern USPs’ self-assembly into nanofibers and 3D networks and how these supramolecular features translate into key rheological properties such as shear-thinning, rapid gelation, and mechanical tunability. We then survey reported applications in tissue engineering, wound healing, and organotypic models, as well as emerging ultra-short peptide-based systems for drug delivery, biosensing, and imaging, highlighting examples where printed constructs support cell viability, differentiation, and matrix deposition. Attention is given to hybrid and multi-material formulations in which USPs provide bioactivity while complementary components contribute structural robustness or additional functionality. Finally, this review outlines the main challenges that currently limit widespread adoption, including achieving high print fidelity with cytocompatible crosslinking, controlling batch-to-batch variability, and addressing the scalability, cost, and sustainability of peptide manufacturing. We conclude by discussing future opportunities such as AI-assisted peptide design, adaptive and multi-material bioprinting workflows, and greener synthetic routes, which together may accelerate the translation of ultra-short peptide-based bioinks from proof-of-concept studies to clinically and industrially relevant platforms.

## 1. Introduction

Peptides are valued in biomedicine for their ability to self-assemble into various molecular forms. USPs, with fewer than eight amino acids, have gained particular attention [1,2,3]. The relatively short chain allows for ease in manipulation, which in turn can alter properties of the peptide itself. Mechanical properties such as stiffness, elasticity, and strength can be adjusted by manipulating the amino acid sequence [3]. As with short peptides, USPs have the capacity for self-assembly, a quality that is highly invaluable for the biomedical world. These peptides can form various structures using this affinity for self-assembly. Adding to the strength of ultra-short self-assembling peptides is the ease of reproducibility, biodegradability, and biocompatibility. Their cost-effectiveness is also an appealing factor that cannot be ignored. These additional properties have further pushed USPs into the spotlight for use in clinical settings [3,4].

As researchers became more acquainted with USPs, their potential uses in 3D bioprinting have not gone unnoticed. Three-dimensional bioprinting itself proved to be revolutionary for the biomedical industry; it is a technology that uses the concept of depositing material such as bioinks layer-by-layer to make a 3D structure [5]. This process can be precisely controlled through computer software, with adjustments having the ability to be quickly made [5]. Three-dimensional bioprinting technology has made rapid advancements throughout the years, with revelations being made in the field of tissue engineering [5,6,7]. The development of organs, blood vessels, bone, and skin, amongst other tissues, is currently being investigated using bioprinting [8]. In these fields, self-assembling peptides have been widely used with reported success [8,9]. Additionally, 3D bioprinting itself can be cost-effective in terms of software and may be more easily accessed as a result [10].

One of 3D bioprinting’s most trying challenges is finding a suitable bioink that can serve its many uses [11,12]. In this area, the unique properties and abilities of USPs may be of use [13]. In this review, we explore and highlight the synergy of USPs and 3D bioprinting.

## 2. Properties of Ultra-Short Peptides

### 2.1. Structural Simplicity and Self-Assembly

The discovery of self-assembly in some peptides has been a revelation for several medical studies. Self-assembly is the ability of biological units to form into structures without the aid of external forces [1]. USPs’ aptitude for self-assembly lends itself well to forming several molecular architectures. These processes are driven by noncovalent interactions such as pi–pi stacking, hydrogen bonds, Van-der Waals forces, metal coordination, and hydrophobic interactions [2,9,14]. It can also be of note that protecting groups such as fluorenylmethyloxycarbonyl (Fmoc) can affect the self-assembly of peptides [15]. The sum of these interactions gives rise to more ordered structures with hydrogen bonding being a determinant of the peptide’s secondary structure (α-helixes and β-sheets) [1,2,9,15]. Self-assembly itself provides a bottom-up design that allows for customization on many levels [1,9]. What is more is that the self-assembly process can be regulated via extrinsic (temperature and pH) and intrinsic (peptide concentration and peptide sequence) factors, allowing it to suit further needs [9,16,17].

#### 2.1.1. Nanostructures

USPs can assemble into nanostructures (Figure 1) such as nanotubules, nanofibrils, nanotapes, nanodoughnuts, and nanospheres, among others [2,18]. These structures resulted from the peptide sequence’s amphiphilicity or through self-complementarity [15]. The nanofibers are of note. These nanofibers can aggregate together to form hydrogels, which can be used to suit a variety of applications [19]. Singh et al. found that several novel nanoarchitectures (nanoweb and nano-saturn) could be assembled from the rearrangement of functional groups and the introduction of metal ions [20]. This suggests that there may be other nanoarchitectures that can be achieved by introducing other components.

#### 2.1.2. Hydrogels

USPs could form hydrogels among other structures [4,15]. These 3D hydrogel structures are reminiscent of the extracellular matrix (ECM) and hold vast amounts of water [9,15,19]. The formation of hydrogels is most often caused by extrinsic factors like changes in temperature, pH, or salt concentrations [16,17]. The mechanical properties of these hydrogels can be adjusted with ease by adjusting the peptide concentration, which further aids in customizability [21]. It is also possible to adjust other properties through the modification of the peptide structure [1].

### 2.2. Biocompatibility and Biodegradability

The self-assembly of USPs leads to great usage in the biomedical field due to their high biocompatibility and biodegradability [15,22]. They are used as vessels for drug delivery, tissue engineering, and biosensors largely thanks to their biocompatibility [1,9,23,24]. Their biodegradability is of note as well for these purposes. In drug delivery systems and 3D printing, being appropriately biodegradable allows for proper function depending on the goal at hand [15,21,25]. Ma et al. observed effective dipeptide delivery carriers that successfully administered antitumor therapy [26]. In this study, the dipeptide, diphenylalanine cationic peptide (CDP, H-Phe-Phe-NH_2_ HCl) was used as an injectable drug carrier in mice. The drug therapy showed significant tumor inhibition in comparison to the control group. The success was owed in part to the biodegradability of the peptides themselves, as there was no negative immune response observed in the mice organs. Furthermore, consistently in many studies, USPs have shown to have a low cytotoxicity and thus could be safe to use in medical applications [27,28,29]. Thota et al. constructed a dipeptide USP hydrogel with Leucine-α, β-dehydrophenylalanine (LeuΔPhe) [29]. This hydrogel they developed was highly successful in drug delivery. It greatly controlled the tumor growth present in mice while strengthening the administered antitumor drug. Thota et al. reported that the gel had high biocompatibility, mechanical strength, and injectability with proteolytic stability [29]. Additionally, there was no detected cytotoxicity as measured by the LDH assay. These are just a couple shown instances of their advantages in biomedical applications as a result of their biocompatibility and biodegradability.

### 2.3. Functional Versatility

The short amino acid sequences of USPs can be manipulated in ways such that the material properties are altered. Modifications have been made to the sequence, backbone, and sidechains. These modifications mean that there can be many customizations made to suit any need [1,3]. For example, if a peptide hydrogel was lacking in mechanical strength, this may be remedied by introducing a urea modification into the structure [30]. The addition of aromatic rings in the center of a peptide chain can improve gelation [1]. Additionally, N and C terminal modifications are a possible method of adjusting the qualities of a peptide. Depending on the exact change to these terminals, the peptide’s hydrophobicity, stability, and rigidity can be modified [1,31]. New attributes, such as antimicrobial properties, may be introduced into the USP system, moreover [3]. With the backbone of the peptide, alterations affect the stability of the system along with its stiffness and ability to self-assemble [3]. All these areas for modification offer many possibilities to create a peptide well-suited for biomedical applications.

### 2.4. Material Strength and Stability

Material strength and stability are essential qualities needed to smoothly carry out a few functions. Mishra et al. conducted research that the majority of USPs hydrogels had high mechanical strength [19]. When compared to collagen gels, the USPs hydrogel used outperformed it in strength. In terms of stability, USPs hydrogels are noted to be stable against physical temperature changes and enzymatic degradation. The stable structure of these peptides is a result of hydrogen bonding and can hold up to the stability of longer peptides [15]. USPs have consistently demonstrated these properties in various studies. One such instance is when a USP hydrogel was used to aid in the healing of rabbit bone defects. The tetrapeptide, Asp–Leu–IIe–IIe, formed a hydrogel capable of supporting the growth of osteoblast-like cells. The bone densities of the defected zones were found to be similar to areas with no defects at all when in the treatment group. The hydrogel was able to support osteoblast differentiation, all while maintaining high cell viability. The hydrogel itself was high in mechanical strength and held against temperatures up to 80 °C. The strength of the hydrogel had a positive correlation with higher peptide concentrations due to thicker fibril meshes being formed. Though even at lower peptide concentrations, the mechanical stability of the peptide was high [28].

In summary, the mechanical properties of USP hydrogels are a direct result of the interaction among peptide sequence, supramolecular organization, and environmental parameters. Aromatic residues and urea or carbamate moieties enhance π–π stacking and hydrogen-bonding density, typically leading to higher storage modulus and better shape fidelity under load. Sequences with less hydrophobic content or weaker interactions create softer, easily extruded networks but may lack long-term stability. Adjusting peptide concentration, ionic strength, and terminal modifications enables precise engineering of USP hydrogels to meet 3D bioprinting requirements.

### 2.5. Interactions with Other Materials

Peptides have been shown to be compatible when combined with other materials such as metals and polymers. Metals have been found to assist with the self-assembly of peptides and can trigger the gelation process [16]. The introduction of metal cations has also been shown to also have effects on the mechanical properties of their resulting hydrogels. Using a pyrene-peptide except for Cu+, divalent and trivalent cations produced stiffer gels [16]. pH changes can trigger self-assembly and also influence the mechanical properties of peptide hydrogels [17]. Chen et al. found that high pHs in combination with metal salts can produce stiffer, less viscous gels [17]. Another study conducted by Saeed et al. successfully combined silver with a tripeptide to form an antibacterial nanohybrid [32]. USPs can change mechanical properties and even show antibacterial properties due to interactions with metals.

Polymer interactions with USPs have been investigated in the form of nanoparticles. Criado-Gonzalez et al. observed how the interaction between these two materials would yield properties that are beneficial in treating inflammatory pathologies [33]. A hydrogel consisting of a phosphorylated tripeptide (Fmoc-FFpY) and polymer nanoparticles (poly(HKT-co-VI) NPs) was produced and exhibited fast gelation due to the electrostatic interactions between them. The resulting gel was self-healing, biocompatible, and showed anti-inflammatory properties. The anti-inflammatory benefits were achieved due to the nanoparticles inhibiting macrophage polarization and decreasing the production of nitrous oxide as a result [33]. Another USP polymer hybrid, poly(N-acryloyl-L-phenylalanyl-L-phenylalanine methyl ester) (polyA-FF-ME) nanoparticles, had the ability to inhibit the Aβ40 fibrillation process [34]. The hydrophobic intermolecular interactions between the peptide residues aided the nanoparticle in disrupting Aβ40 fibril assembly, as well as increasing the fibrillation lag time [34]. The synergy of USPs and polymers has already yielded possible applications in biomedical applications. Using the interactions between these biomaterials can be of use in discovering further treatments.

## 3. Synergies Between USPs and 3D Printing

### 3.1. Why USPs for 3D Printing?

USPs have shown great promise for use in 3D printing. Most of the research revolves around using hydrogels as a biomaterial for printing. Other natural bioinks face the issue of having weak shape fidelity, low biocompatibility, and the use of harsh chemicals or harmful crosslinking [14,35]. Furthermore, the addition of cells after the initial printing can be difficult when using other natural bioinks [35]. USPs have the potential to overcome these challenges through their unique properties.

To serve as a bioink, the hydrogel should be shear-thinning, rapidly gel, and exhibit minimal swelling after extrusion [36]. Macroscopic properties stem from sequence-level design. Increasing aromatic or bulky hydrophobic residues enhances π–π and hydrophobic interactions, resulting in denser fiber networks and higher yield stress, which improves filament integrity. Adding charged or hydrophilic residues lowers viscosity and aids shear-thinning but may reduce stiffness if not balanced. This molecular tunability enables bioinks to meet the rheological requirements for effective 3D bioprinting. Furthermore, adequate biocompatibility must be taken into consideration so that cells are able to be supported and can maintain function once printed [9]. Considering that hydrogels demonstrate a high level of customizability, they can be adjusted to meet all the criteria of an effective bioink. Tetrapeptide hydrogels were tested to address common bioink challenges and assess their suitability for cell support [35]. Not only did the peptide bioinks successfully print more complex structures such as noses, but they were able to maintain its shape integrity over several weeks, all while supporting a cellular system. Susapto et al. found that their USPs could encourage the growth of mouse cortical neurons [35]. In addition, this same study was able to have its printed scaffold further mimic functional tissue by inducing cell differentiation. Jian et al. reported that adjusting peptide concentration allows for tuning the mechanical properties and biodegradability of Fmoc-YD and Fmoc-YK dipeptide hydrogels. While bioink construct shape fidelity remains challenging, it may be improved by optimizing peptide concentrations and mixing ratios. Notably, combining Fmoc-YK and Fmoc-YD increased system strength by over fivefold compared to each peptide alone. The studies by Susapto et al. and Jian et al. show that minor changes in peptide sequence or formulation can significantly impact nanofiber density, network connectivity, and thus gel properties like stiffness, degradation, and shape fidelity. This demonstrates that USP bioinks can be tailored at the sequence level for specific applications [21,35].

### 3.2. Current Methods Integrating USPs

Several 3D printing methods have been developed, with extrusion and inkjet techniques being some of the most prevalent used. Extrusion techniques rely on the dispensing of bioinks through a nozzle via air pressure pneumatic systems or mechanical systems. The resulting product is a cylindrical filament that can be layered to make 3D constructions [7]. With extrusion-based printing, there are three mechanisms: piston, screw, and pneumatic [5,7,37]. Using this method requires that the bioink has a high viscosity for more supported structures but not so high that the nozzle clogs [5,37]. It can be noted that this method may cause cell damage due to small nozzles or too much pressure [5,38].

Inkjet techniques utilize pressure on the printhead to produce droplets of hydrogel [35]. This method notably produces a construct with high cell viability and high print resolution. The bioink for inkjet techniques requires low viscosity to work properly. Drop-on-demand (DOD) and continuous inkjet are the two categories of inkjet printing. Continuous inkjet is not currently used for bioprinting due to unavoidable ink contamination, leaving DOD as the choice for inkjet printing. DOD can be further separated into thermal, piezoelectric, and electrostatic inkjet techniques based on how the bioink is ejected from the nozzle. Inkjet printing provides high resolution but may be limited to smaller constructs compared to extrusion printing [7,39].

Another technique to mention is laser-based bioprinting. While it may not be as prevalent as the other two methods, it still has a place in peptide bioprinting. This technique does not require nozzles and instead uses lasers to deposit bioink [5,7]. Laser-based printing typically has a combination of a laser and a mirror or lens. The laser is directed via the lens/mirror towards the bioink, causing high gas pressure followed by the formation of air bubbles. These bubbles are the driving force behind the deposition of the bioink droplets [5]. Stereolithography is a laser-based printing method that utilizes UV light for curing the construct layer by layer. It is advantageous for high-resolution prints, which is a stated advantage for laser-based printing methods [7]. However, laser-based printing is reported to be expensive and so may limit use [5].

### 3.3. Challenges and Opportunities

Three-dimensional bioprinting comes with its own set of challenges to be effective in its applications. Ensuring that the fidelity of the 3D constructs is high remains one of its more daunting challenges [35]. Another issue comes with curing the printed constructs [40]. The issue of curing revolves around keeping the cytocompatibility of the construction while simultaneously strengthening it [40]. Curing via photo-cross-linking is a notable technique often used. The intensity of light may be too harsh and thus exhibit cytotoxicity and make it unsuitable for many biomedical uses [41]. To advance this technology, curing methods must maintain the biocompatibility of hydrogel bioinks [42]. This would strengthen constructs and expand bioprinting possibilities.

It was noted that when using hydrogels, viscosity was a major factor in determining their printability and cell encapsulation [10,43,44]. Adjustable viscosity enables varied printing techniques and maintains high print quality [35,45]. Although peptide hydrogel viscosity is modifiable, selecting materials and designing scaffolds are still challenging tasks [6].

With the challenges that come with using USPs, there are also opportunities that present themselves. Seeing as there are so few amino acids in each sequence, this means that the cost of synthesis is lower compared to peptides with longer sequences [46]. This would make experimentation more accessible for researchers, as well as allowing it to be a candidate for mass production. Moreover, there has been ample research demonstrating the use of USPs in 3D bioprinting. Susapto et al. designed self-assembling tetrapeptides, Ac-Ile- Ile-Phe-Lys-NH_2_ (IIFK), Ac-Ile-Ile-Cha-Lys-NH_2_ (IIZK), and Ac-Ile-Cha-Cha-Lys-NH_2_ (IZZK), for the purpose of being bioinks [35]. The study demonstrated that USPs are a promising biomaterial: The ink was biocompatible, retained shape fidelity, and supported cell growth. USPs address several limitations of other bioinks, such as harsh cell conditions and harmful crosslinking [35]. While some challenges remain, Susapto et al.’s research suggests that USPs have the potential to overcome them.

## 4. Applications for USPs in 3D Printing

### 4.1. Biomedical Applications

At present, scientists focus on USPs usage for biomedical applications since their cost-effectiveness, simple development, and their ability to be produced on a large scale categorize them as fascinating biomedical systems. Numerous peptides have shown significant biological roles as active molecules. This includes immune defense enhancement, anti-tumor therapy, metabolic syndrome, and bone tissue engineering [47]. Over the years, USPs have been progressively developed into novel biomaterials for a range of applications. There is much intrigue in USPs ability to form 3D networks as hydrogels [48]. This potential leaves room for USPs to be utilized as probes for bioimaging [49], inks for bioprinting [48], tissue engineering scaffolds [50], and drug vehicles [51]. And as previously discussed, USPs are capable of self-assembling into well-ordered nanoplatforms, i.e., nanoparticles [52], nanofibers [53], or nanotubes [54]. Figure 2 summarizes the process from peptide design and self-assembly to hydrogel bioink formation and 3D bioprinting.

#### 4.1.1. USP-Based Scaffolds for Tissue Engineering

Three-dimensional printed scaffolds fabricated using USPs technology can replicate the ECM, thereby facilitating cell adhesion, proliferation, and differentiation. Such scaffolds are utilized in engineering cartilage, bone, and neural tissues. To facilitate comparison across studies, Table 1 summarizes representative USP-based bioinks, their printing modality/printability, and reported cell viability/biological outcomes.

According to Loo et al., ultrashort peptide hydrogels were utilized as a bioink to fabricate hydrogel scaffolds through 3D bioprinting. The USPs can instantly gel under physiological conditions to form peptide-hydrogel scaffolds. Such gel scaffolds then facilitate the growth of primary cells while encapsulating human stem cells. Through the bioprinted scaffolds, the cells underwent differentiation, leading to the formation of primary cells into skin as well as intestinal organotypic tissue constructs. Therefore, they can be a useful platform for diagnostics and drug screening. In theory, di-, tri-, and tetra-peptides could also be employed as bioinks besides hexapeptides, which were used by Loo et al. [55]. Arab et al. designed hydrogel-based scaffolds based on novel tetramer peptide biomaterials. Two CH-01 and CH-02 USPs, which can self-assemble nanofibrous 3D networks, were studied; these were able to entrap aqueous media, assimilating the native collagen of the extracellular matrix (ECM). Besides their biocompatibility, which was tested by various assays on mouse myoblast cells (C2C12), the peptides showed great printability. Authors have examined them as peptide-based bioinks by utilizing a market-available extrusion-based 3D bioprinter. It was concluded that USPs have a viable application as 3D bioprinted scaffolds carrying skeletal muscle myoblasts [56]. The same group expanded the knowledge on the bioprinting of the USPs (CH-01 and CH-02), given their good printability and custom-designed 3D bioprinted scaffolds, as well as their property to mimic ECM complex structures. The researchers applied the robotic 3D bioprinter to develop the scaffolds. Considering the live–dead assay findings, it was revealed that the 3D-printed scaffolds were able to improve adhesion and proliferation for a minimum of five days. Moreover, according to the 3D culture, USP-based scaffolds promote conversion of C2C12 cells into muscle fibers. Therefore, the 3D bioprinted biomaterials based on CH-01 and CH-02 can be applied for skeletal tissue engineering and regeneration fields [57].

Susapto et al. outlined the design method of potential bioinks for automated complex tissue fabrication; the bioinks derived from novel aromatic and non-aromatic tetrapeptide amphiphiles, IIFK, IIZK, and IZZK. It was revealed that the used USPs could shape into transparent hydrogels even at 0.1% *w*/*v* and desirable mechanical stiffness suitable for extrusion bioprinters. At physiological conditions and low concentrations that are considered cost-effective, peptide bioinks fostered instant solidification during the 3D printing process. The fabrication of tissue scaffolds and the structural integrity of cells over a period of weeks, even though the strenuous process of printing, was confirmed. Moreover, human mesenchymal stem cells containing the bioink were also tested for chondrogenic development, showing that peptide bioinks have great potential for the automated construction of complex tissue [35]. In another study, Alhattab et al. [14] constructed amphiphilic tetrapeptides composed of a hydrophobic tail and a cationic head group, which can self-assemble to nanofibrous hydrogels under physiological conditions. The two promising USPs, IIZK and IZZK, were studied as bioinks for in vivo cartilage fabrication due to their biocompatibility and human mesenchymal stem cells (hMSC) differentiation to chondrocytes.

Rauf et al. developed Ac-Ile-Val-Cha-Lys-NH_2_ (IVZK) and Ac-Ile-Val-Phe-Lys-NH_2_ (IVFK) USPs and studied their printability via an in situ 3D bioprinting method. Their method is described as more advantageous compared to other printing methods since the scaffolds can be printed under truly physiological conditions. Additionally, the IVZK and IVFK bioinks showed durability and printability, offering great biocompatibility when studied on human fibroblasts and (hMSC). The authors generally concluded that by improving bioinks and printing methodology, the 3D bioprinting field will benefit, along with the fields concerning tissue engineering and regenerative medicine [58]. Khan et al. [43] reported the printability of two tetrameric USPs bioinks, as also described by Rauf et al. for cartilage tissue engineering applications. The authors found that adding a vacuum system to the robotic 3D bioprinter improved scaffold printability and resulted in finer peptide hydrogel patterns. Moreover, 40 mm cylindrical constructs with less water content were created in their study, allowing the structure to stay firmly in place. Another study aiming to develop suitable 3D printing biomaterials for cartilage tissue regeneration was performed by Ahn et al. [59] The authors used poly(propylene fumarate) (PPF) materials as bioinks for 3D printing in the production of macroporous cell scaffolds by micro-stereolithography. They further immobilized arginine–glycine–aspartate (RGD) peptides so as to enhance the cell–matrix interaction of human chondrocytes in the 3D scaffolds. The 3D bioprintable scaffold may encourage the regeneration of cartilage tissue because of human chondrocytes’ adhesion and proliferation being efficiently supported.

Design-wise, these examples show how minor amino acid changes lead to different mechanical and printing properties. In CH-01 and CH-02, a balance of hydrophobic and charged residues creates ECM-like nanofibrous networks that support myoblast adhesion and differentiation. For tetrapeptides like IIFK, IIZK, IZZK, IVZK, and IVFK, the hydrophobic tail and Lys headgroup determine amphiphilicity and packing, resulting in hydrogels with shear-thinning and quick recovery after extrusion. Understanding these structure–property relationships is key to optimizing USPs as bioinks to enhance printability, shape fidelity, and cell support in tissue engineering.

#### 4.1.2. USPs for Drug Delivery Systems

It can be hypothesized that drugs have the potential to be released from 3D-printed USP matrices in a controllable manner; accurate drug delivery occurs in accordance with degradation ability in the physiological environment. Despite this, limited applications of 3D printing can be found and mostly examine the release of USPs from 3D-based systems or other systems. Emtiazi et al. applied diphenylalanine peptides (FNTs) functionalized with folic acid/magnetic nanoparticles and evaluated them as an anti-cancer drug delivery system of 5-fluorouracil (FU). According to in vitro release findings, the FU is released within four hours with a significantly reduced release rate [60]. Moreover, another research group analyzed the direct conjugation of 5-Fu to Di lysine peptides, which were self-assembled into different nano-morphologies, including nanotubes, nanofibers, and nanobelts. Although these studies do not involve 3D printing, they can provide a basis for future studies [61]. Lim et al. studied the transdermal delivery of acetyl-hexapeptide 3 (AHP-3), which is known for its antiaging abilities. Despite its efficiency and safety, AHP-3 demonstrated low permeation because of its high molecular weight and hydrophilicity. AHP-3 release took place through its incorporation into a 3D-printed microneedle (MN) system composed of poly(ethylene glycol diacrylate) and vinyl pyrrolidone. The personalized MNs were developed via digital light processing after employing software of computer-aided design. The results revealed that the cytocompatible MNs-based patch can remain stable after compression while the peptide successfully penetrates human cadaver dermatomed skin [62]. Figure 3 exhibits various photos of the 3D-printed MN patch with AHP-3. In general, short peptides have been incorporated into microneedle-based systems, either prepared by 3D printing or other techniques and examined for their properties [63].

#### 4.1.3. USPs for Biosensors and Imaging Probes

It has been reported that functionalized USPs can be integrated into 3D-printed biosensors that are able to detect biomolecules, such as glucose or pathogens, with high selectivity. Nanoparticles from USPs as tryptophan–phenylalanine (DNPs), have the capacity to shift the peptide’s intrinsic fluorescence from ultraviolet into visible wavelengths. Because of their visible emission signal, these nanoparticles were studied as imaging and sensing probes and drug delivery vehicles of doxorubicin by Fan and coworkers [64]. It was shown that the DNPs presented visible fluorescence, a narrow emission bandwidth, photostability, and biocompatibility. The functionalized DNPs with doxorubicin and MUC1 aptamer can target cancer cells for imaging and monitoring delivery in real time. Almohammed et al. utilize 3D printing for surface-enhanced Raman spectroscopy (SERS)-based sensing applications. SERS templates are frequently used as sensors due to their high sensitivity and reproducibility, as well as cost-effectiveness and ease of preparation. The 3D-printed SERS templates were fabricated utilizing a Fmoc-Phe-Phe hydrogel incorporating silver or gold nanoparticles. The sensor was able to detect adenine and other low Raman cross-section molecules with concentrations lower than 100 pM [65].

### 4.2. Other Applications of 3D-Printed USP Hydrogels

USPs can be engineered to form optically or electrically active materials, enabling the production of conductive inks for flexible electronics, photonic materials for light manipulation, as well as piezoelectric materials for energy harvesting. Handelman et al. examined linear aromatic diphenylalanine, cyclic diphenylalanine, linear aliphatic dileucine, and linear triphenylalanine, which can self-assemble into hollow peptide nanotubes, nanospheres as well as nanofibers. The supramolecular structures can irreversibly transform through exposure at increased temperatures ranging from 140 to 180 °C. The USPs can reassemble into a phase with thermodynamic stability, mimicking the nanowire amyloid fibrils. Moreover, these USPs exhibit photoluminescence. It can be concluded that the USPs can act as self-assembled dyes being used as intrinsic optical labels in biomedical microscopy as well as biolasers and biocompatible markers [66]. A future study can involve the 3D printing of these USPs and their examination for various advanced applications. Yang et al. [67] combined the freeform, meniscus-guided 3D printing with molecular self-assembly to fabricate crystalline diphenylalanine (FF) peptide platforms. The findings revealed that thanks to its crystallinity, 3D-printed FF depicts piezoelectricity that can be applied to bioelectronic devices. Safaryan et al. investigated the controlled deposition of FF through a commercial inkjet printer. The original FF solution was modified in such a way so as to be utilized as an efficacious ink for the printing of aligned FF structures. In addition, authors created FF structures with exceptional stability and increased piezoelectric responsiveness during the ink development process [68].

## 5. Future Perspectives

### 5.1. Use of Artificial Intelligence to Design Peptides for Specific 3D Printing Applications

The merging of USPs and 3D printing technologies presents an enticing future scenario for the engineering of advanced functional biomaterials. However, at the turn of the new era, integrating artificial intelligence (AI) to optimize peptide design, performance, and application on 3D-printed constructs is set to become the next big frontier. AI provides all the tools needed to confront the countless intricacies of these peptide-based material systems, allowing the prediction, generation, and refinement of USPs according to pre-established structural, mechanical, and biological grounds. Designing USPs for tunable self-assembly, mechanical integrity, and biological function presents a staggering amino acid combination sequence space. These are typically designed through slow and tedious experimental screening or rational design. AI, especially through generative models, including recurrent neural networks (RNNs) and transformer architectures, can be used for in silico generation of peptide sequences with well-defined biophysical properties such as hydrophobicity, charge distribution, and the propensity to form β-sheets or α-helices. Generative adversarial networks (GANs) and variational autoencoders (VAEs) have been successfully used for de novo peptide sequence design, creating novel bioactive peptides not annotated in common natural product databases. Such methods can be further harnessed for the generation of new USP candidates, which can then be used for specific functions in 3D printing, such as rheology tuning for extrusion, hydrogelation behavior, or scaffold stiffness, by training on curated datasets of functional peptides [69].

AI models can also predict the 3D conformation and assembly behavior of peptide sequences—a crucial parameter for 3D printing, where the supramolecular architecture determines mechanical performance and print fidelity. For instance, supervised learning algorithms can work with structural data of known peptides to predict self-assembly pathways or gelation propensity. Alongside molecular dynamics simulations, these predictions assist in determining the stability of peptide aggregates under extrusion pressures or thermal gradients observed during printing. Reinforcement learning, a subset of artificial intelligence in which systems refine strategies through iterative feedback, can be employed to optimize peptide sequences for multifunctionality—such as combining antimicrobial properties with enhanced mechanical strength. This approach to multi-objective optimization is particularly valuable in additive manufacturing applications that demand constructs that are both bioactive and physically robust [70].

Perhaps, among the most disruptive implications of AI in peptide-based 3D printing is QbD, wherein the inter-relationships among critical material attributes (CMAs), critical process parameters (CPPs), and critical quality attributes (CQAs) may be both modeled and controlled. AI models could be used to determine which USP sequences would best satisfy demanding criteria related to printability, structural fidelity, degradation rate, and biological compatibility, all useful for biomedical applications such as tissue engineering or in vivo implants [71]. This AI-augmented multi-modal sensing system provides for data-driven design with real-time feedback, such as viscosity, shear-thinning behavior, and cross-linking kinetics being measured during the printing process for in-line system adjustments to formulation or printing parameters. These closed-loop (QbD) workflows reduce reliance on trial-and-error optimization and can improve reproducibility and scalability of printed constructs [71].

AI can assist in promoting USPs into 4D printing, wherein materials change shape or function as a function of time in response to environmental stimuli. AI can identify motifs or structural inducers that impart specific transformations, like pH- or temperature-induced folding, by the extraction of vast data sets of responsive peptides. These characteristics can be incorporated within USPs for the design of dynamic scaffolds for several applications, like targeted drug delivery, wound healing, and regenerative medicine, as shown in Figure 4. Furthermore, the process of integrating stimulated peptides into complex architectures can be automated through AI tools to ensure that the spatial arrangements in a 3D-printed object allow for a coordinated response. Such advanced predictive and spatially aware control is largely unobtainable through a traditional design pipeline, thus marking the importance of AI in 4D peptide-based manufacturing [72], as shown in Figure 4.

An exciting emerging research area is the AI-guided in situ bioprinting, where the systems adapt printing behavior dynamically according to live information collected from imaging, patient anatomy, or biological cues. Surgical robots could conceivably print peptide-based materials directly onto the tissues during surgery in an AI-guided approach, dynamically modifying layer geometry or composition in response to the biological milieu. In this context, AI could gather real-time mechanical constraints, physiological information, and structural needs to engineer and deposit peptides in situ [73]. Being a real-time and adaptive printing paradigm, novel clinical workflows may be exploited wherein personalized peptide-based constructs are fabricated intraoperatively, with utmost precision to the tissue geometry and patient-specific cues. Therefore, AI becomes not only an intermediary design tool but also an active stakeholder in the production chain.

Yet the promise was still shackled with certain bottlenecks that formerly interfered with AI peptide design. A lack of sufficiently large, sufficiently informative databases for short peptides, especially USPs with relevant parameters for printing (e.g., mechanical strength and gelation kinetics), hinders model training. A request comes to public repositories and collaborative initiatives to fortify databases with experimentally validated peptide sequences and functional annotations. Secondly, interpretability of modeling constitutes another bottleneck; although highly effective, deep learning systems usually execute their task like a “black box”. For regulatory acceptance and eventually biological insight, more transparent AI models and explainable AI frameworks are needed. Third, most of them have yet to be standardized to be used as a reference point to enable comparison among various AI-designed peptide candidates for printability, self-assembly, and functional performance [74]. To accelerate innovation, collaborative platforms are needed that integrate automated synthesis and testing systems with computational design. For 3D bioprinting, future USP datasets should include not just sequence and bioactivity data but also standardized rheological and printability metrics like gelation kinetics, yield stress, shear-thinning, filament stability, and long-term durability under physiological conditions. Without unified protocols and benchmarks, AI models risk overfitting to narrow lab conditions instead of yielding broadly useful design rules. As AI-designed peptides approach clinical use, ethical considerations and regulations are essential to ensure safety, immunogenicity, and lasting effects, all under strict and controlled evaluation.

### 5.2. Emerging Technologies

#### 5.2.1. Hybrid Materials Combining USPs with Other Advanced Materials

The hybridization of USPs with cutting-edge, high-tech materials reveals new possibilities for the design of multipurpose materials based on biotechnological and 3D printing applications. Through the integration of polymers, inorganic nanomaterials, and intelligent biomaterials with the molecular precision, self-assembly ability, and biocompatibility of USPs, hybrid systems offer improved mechanical properties, bio-functionality, and stimuli-responsiveness. To facilitate the synthesis of strong composite structures with improved material properties, hybrid systems take advantage of specially designed molecular-level USP–material interactions, including ionic interaction, covalent bonding, hydrogen bonding, or π–π stacking. For example, incorporation of USPs in polymeric or inorganic scaffolds enables hierarchical self-assembly into fibrous, porous, or crystalline networks for tissue scaffolding and bio-fabrication [75,76]. Peptide and polymer conjugates make mechanical and biological action controllable, such as in the case of elastin-like peptide hydrogels or PEGylated USPs. Hybrid systems have the capability of healing and instructing cells, the actual needs of regenerative medicine [77].

A key drawback of USP-only systems is their comparatively weak mechanical strength. This issue can be addressed by incorporating nanostructured materials such as carbon nanotubes, graphene oxide, or silica nanoparticles. These additives significantly improve the system’s tensile strength, elasticity, and stability without compromising the bio-functionality provided by the peptide components. For instance, Li and colleagues developed a bioink combining peptide nanofibers with inorganic nanosilicates, which exhibited excellent shear-thinning behavior and shape recovery during extrusion-based printing. This facilitated the creation of cell-laden structures with high viability [76].

USPs of responsive polymers, such as PNIPAM, poly(acrylic acid), or polydopamine, facilitate environmental monitoring and dynamic functional response to stimuli such as temperature, pH, light, or oxidative stress. Such systems can gelate, degrade, or swell in a programmed manner well suited for drug delivery or cell encapsulation [78]. These systems also allow post-printing functionalization; for example, USP–polydopamine composites can easily immobilize growth factors, antimicrobial agents, or imaging molecules for theranostic use [77].

Combining USPs with biocompatible polymers resulted in highly printable hydrogels with shape fidelity and mechanical integrity throughout and after printing. These hydrogels can be loaded with various materials, such as gelatin methacrylate (GelMA), alginate, or cellulose nanofibers, to maintain rheological properties with maintained cell viability and support. Extruded, light-guided, and inkjet 3D printing-compatible hybrid peptide–polymer systems have been shown in recent research. These bioinks can be derived and engineered for a specific application—bone repair, tissue repair, or organ-on-chip models—by modulation of peptide sequence and polymer structure [24,79].

USPs can also be blended with inorganic phases like hydroxyapatite, bioactive glass, or metal–organic frameworks to engineer biomimetic composites for tissue engineering with load-bearing functions. While these hybrids are stronger mechanically, they also induce biological responses like osteogenesis or angiogenesis. As an example, a peptide–calcium phosphate hybrid was employed to 3D print scaffolds replicating the mineralized matrix of bone and exhibited enhanced compressive modulus and cell infiltration [80]. In another study, gold nanoparticle incorporation into peptide amphiphile gels was introduced, giving plasmonic functionality to realize photothermal activation for on-demand drug delivery [76].

USP incorporation in hybrid systems allows for the development of “smart” materials that can be triggered to degrade, self-report, or elicit stimuli upon exposure. ROS-responsive peptide–polymer hybrids, for instance, have been investigated for drug delivery in inflammatory conditions, and redox-responsive materials provide controlled release in cancer cells [76,77]. Hybrid peptides integrated into electroactive or fluorescent matrices also provide biosensing, wherein the peptide provides molecular recognition, and the hybrid matrix provides signal transduction.

According to the circular economy concept, recent hybrid systems also focus on sustainability. Agri-marine biomass-derived natural peptides can be blended with renewable resource-based biodegradable polymers like PLA, PHA, or chitosan. Hybrid products are in the spotlight for green packaging, bioelectronics, and environmental sensors [79].

There remains a challenge despite the swift progress pace. There is control over phase separation in the nano–bio interface, homogeneous hybrid filler dispersion, and scale-up synthesis at the expense of neither reproducibility nor safety. Other regulatory hurdles to the clinical translation of peptide-based hybrid materials remain considerable and comprise extensive biocompatibility and toxicology testing. Nevertheless, larger strides in synthetic biology, supramolecular chemistry, and additive manufacturing will make possible the rational design of multifunctional USP-based hybrid systems customized for use by individuals in medicine, industry, and environmental technology.

#### 5.2.2. Multi-Material 3D Printing Techniques Using USPs

The advent of multi-material 3D printing techniques has drastically extended the limits of additive manufacturing, especially in the synthesis of intricate, heterogeneous, and functional materials for next-generation engineering and biomedicine technologies. In this regard, USPs are attracting more and more attention as multifunctional, bioactive building blocks that may be synthesized into multi-material systems with biological functionality, self-assembly, and molecular tunability. These distinct characteristics enable USPs to become structural and bioactive components in multi-compositional and spatially differentiated printed structures. With multi-ink 3D printing, inks can be deposited concurrently or patterned sequentially to create a sequence of inks with distinct mechanical, biological, or structural uses [81]. This is needed when developing tissue analogs with regionally differentiated properties, such as gradient stiffness or compartmentalized biochemical information. USPs are especially apt for such applications because of the self-assembly of the molecules into nanofibers, hydrogels, or nanostructured assemblies in a sequence-specific manner and conjugation to polymers, nanoparticles, or other biomaterials. These features render them as the most suitable candidates for being integrated into hybrid systems prepared through extrusion-based, inkjet, digital light processing (DLP), or coaxial printing platforms, as shown in Figure 5.

Direct ink writing (DIW) and extrusion-based approaches are some of the most prevalent platforms for peptide-based multi-material bioprinting. In these systems, peptides are generally introduced into carrier hydrogels or nanocomposites, wherein their self-assembling ability plays a role in the structural and bio functional properties of the final construct. Recently, peptide-blended inks have been printed with DIW to create scaffolds with spatially controlled topography and composition to integrate cell-instructive peptide motifs into mechanically strong synthetic materials [83]. The technique is highly intriguing in regenerative medicine, with cell behavior and tissue regeneration being directed by the spatial patterning of biomolecules within scaffolds.

USP incorporation into multi-material printing protocols has also been shown using microfluidics-assisted platforms. Ghalayini et al. employed a flow-focusing microfluidic chip to encapsulate self-assemblages of peptide nanoparticles in hydrogel matrices (during printing) to create spatially resolved composite materials ideal for drug delivery and tissue regeneration [1]. Such a process allowed precise localization of the nanoparticles and their distribution as a key parameter in applications involving targeted therapeutic response or localized cellular signaling.

Coaxial extrusion, another multi-material process susceptible to variability, has been used to co-print peptide hydrogels with reinforcing components like methacrylated gelatin or nanocellulose. In one example, scientists printed concentric core–shell filaments of USP-containing inner layers and photo-crosslinkable polymeric shells. The produced hybrid scaffolds possessed enhanced mechanical properties and shape stability, without compromising the bioactive and cell-adhesive nature of the peptides in the core [84]. This territorial compartmentalization is critical in the establishment of functional gradients within tissues, e.g., between bone and cartilage or within the vasculature.

One of the key issues in co-processing USPs with other materials is rheological and crosslinking compatibility control. USPs in water solutions would otherwise create low-viscosity mixtures that are unsuitable for extrusion in terms of their yield stress. To offset this, the use of viscoelastic modifiers like alginate or gelatin has been employed to increase printability without sacrificing peptide self-assembly. Shahbazi et al. illustrated how the utilization of USP was promoted with thermoresponsive polymers for 4D printing applications, where constructs would change shape under governed environments when exposed to physiological temperatures [83]. This compatibility between USP bioactivity and polymer responsivity presents good potential in creating dynamic implants and adaptive devices.

The main issue of channel crossover and printhead contamination in multi-nozzle systems is among the challenges to using multi-material 3D printing with USPs. One solution to this is the application of controlled valve switching systems or droplet-on-demand inkjet systems that can switch accurately between polymer and peptide inks. During printing, a multi-cartridge inkjet system can deposit various materials in alternation, combining structural resins with peptide-functionalized bioinks in the very same object, making way for controlled heterogeneity within any given individual construct and, hence, steered zoning of differing chemical or biological properties on a microscale [85,86]. Moreover, stereolithography-based and DLP multi-material printing have started to integrate USPs functionalized with photocrosslinkable groups to be selectively polymerized along with synthetic resins. For instance, photopolymerization of acrylated peptides using PEG-diacrylate led to the creation of hydrogels with spatially encoded peptide sequences that guided cell adhesion and migration in a manner unachievable using standard mono-material bioinks [87].

Other than biomedical uses, multi-material printing with USPs has been of interest for soft robotic system design, biosensors, and responsive materials. Combining conductive polymers or metal nanostructures with USPs enabled the creation of soft, bio-interfaced circuits for biochemical signal detection or therapeutic stimulation. Wang et al. have co-printed peptide amphiphile–graphene oxide composites onto elastomeric substrates to create stretchable biosensing platforms [88]. These materials maintain biocompatibility and flexibility while displaying electric responsiveness, proving the versatility of USPs in multifunctional printing systems.

As the maturation of multi-material 3D printing continues to advance, USP design merging with printing hardware, software control, and computational modeling will be crucial. Real-time feedback controls that modulate printing parameters based on peptide rheology or polymerization kinetics can optimize fidelity and function. AI-based design tools will also likely aid in choosing peptide sequences optimized for co-processing, crosslinking compatibility, and synergistic bioactivity. Briefly, USPs and multi-material 3D printing are an active and potent pathway to the construction of bio-relevant, functional, and complex structures. With the ability to control spatial composition at the molecular level and synergistic interactions with other future generations of materials, it is now feasible to create structures that closely resemble the form and function of tissues in vivo but reach into the body for diagnostics, drug delivery, and soft electronics.

### 5.3. Sustainability and Scalability in USP Development

The expanding application of USPs in the biomedical field raised many questions about their sustainability, as well as their scalability. Though USPs possess certain advantages, such as low molecular weight, synthetic tunability, and inherent biodegradability, mass production as well as environmental effects are giving rise to critical concerns. Peptide synthesis has conventionally utilized solid-phase peptide synthesis (SPPS) even though it is extremely efficient and prevalent. But this process is very unfavorable to sustainability because it utilizes a significant amount of toxic solvents, such as N,N-dimethylformamide (DMF) and N-methyl-2-pyrrolidone (NMP), and possesses low atom economy coupling reagents, such as HATU or PyBOP, which are toxic in nature [89].

The environmental and economic cost at the industrial scale is further magnified by the fact that the process mass intensity of peptide synthesis is significantly higher than for small molecules [89]. Additionally, the scalability of USP-derived products is inherently tied to purification methods, which will more than likely encompass large-scale chromatography—yet another energy- and solvent-demanding step [89]. Both of these aspects impede frictionless scaling-up of USPs from research-scale production to clinical-grade manufacture. To respond to such issues, huge strides are being made towards greener and more scalable synthetic methods. Other solvents such as γ-valerolactone and 2-methyltetrahydrofuran (2-MeTHF) are explored as substitutes for SPPS in the direction of reducing reliance on hazardous solvents, and recoverable solid supports are under development to address resin waste [89]. Second, liquid-phase peptide synthesis (LPPS), while more labor-intensive, might possibly be more sustainable for short peptides such as USPs from the viewpoint of low solvent consumption and recycling [90]. On the material level, USPs biocompatibility and biodegradability are a plus over peptidic-free polymers. USPs degrade into amino acids or dipeptides, excluding long-term microplastic problems or toxic degradation products [24]. Enzyme-triggered self-assembly also allows for targeted delivery and local bioactivity, minimizing the needed dose and reducing systemic exposure and therefore indirectly encouraging lower production requirements and waste.

From the green chemistry point of view, scale-up in a sustainable way is also about redesigning upstream building blocks. Chemoenzymatic or biosynthetic transformation of amino acids from renewable raw materials can reduce environmental impact in monomer manufacturing [91]. To decrease batch-to-batch variability and overall efficiency in connection, process intensification methods such as continuous flow peptide synthesis are being examined concurrently [92], as shown in Figure 6.

To pave the way for sustainable development, regulations and industrial infrastructures alike are being modified concurrently. Green innovation in peptide API synthesis has been identified to be needed by institutions such as the ACS Green Chemistry Institute Pharmaceutical Roundtable, which is attempting to eliminate the gap between research development and business application [89]. Synthetic innovation, process development, material life cycle assessment, and cross-sector coordination need to be weighed against each other for sustainable large-scale manufacture of USPs. Such a multi-layered approach will ensure that USPs are incorporated in shared biomedical applications without undermining economic or environmental sustainability.

## 6. Conclusions

Within recent years, USPs have emerged as a popular option for biomedical use due to a number of compelling properties. Their short chain of eight or fewer amino acids lends itself to ease in tuning properties such as strength, rigidity, and stability through various modifications. These peptides have the capacity for self-assembly into nanostructures, which can result in a versatile hydrogel. In 3D bioprinting, USP hydrogels have shown much promise. They have demonstrated potential use in biosensors, tissue engineering, drug delivery systems, electrically active, and optically active materials.

USPs have shown much potential for applications in bioprinting; however, there are present limitations and challenges that must be addressed. Structural integrity is a concern when printing constructs. USPs-only constructs may lack shape fidelity because of low mechanical strength. Often, specific viscosities must be achieved to print structures that maintain shape. Keeping biocompatibility goes hand in hand with this issue. Often, the curing methods for stable constructs can leave them cytotoxic, resulting in unsuitability for certain applications. Scalability for mass production and sustainability are left in question as well due to the use of toxic solvents and energy-demanding purification methods. These are just a few of the hurdles that are present within USP use. Despite this, USPs can have applications in numerous fields, and the possibilities for use can only grow.

There are several avenues for future research involving USP bioprinting. Emerging technologies with AI have the potential to make peptide design and application easier than ever before. Several factors, like USP conformation, behavior, and restraints, can be predicted, leading to the possibility of adaptive printing technology. What is even more exciting is the combination of USPs with other materials. By merging the peptides with different materials, it is possible to address current challenges such as low mechanical strength and shape fidelity. It has been observed within several studies that hybridized materials or multi-material printing can lend abilities to USP systems that would have otherwise been lacking. For example, USP polymer systems can degrade and gel in a controlled manner. With other substances, the resulting USP system can be both mechanically strong and remain biocompatible. Advancements such as these open the door for many more possibilities for USP applications.

## Figures and Tables

**Figure 1 gels-12-00049-f001:**
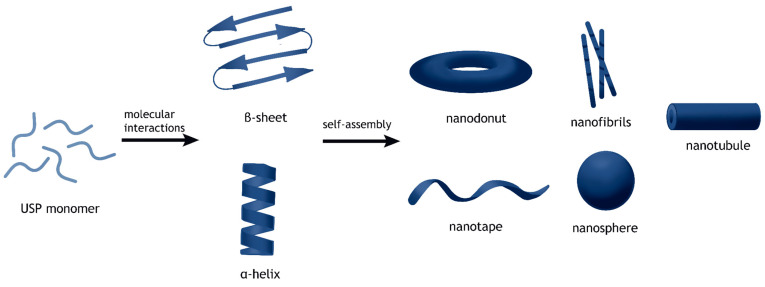
The process of peptide assembly from monomers into secondary structures and secondary structures into nanostructures.

**Figure 2 gels-12-00049-f002:**
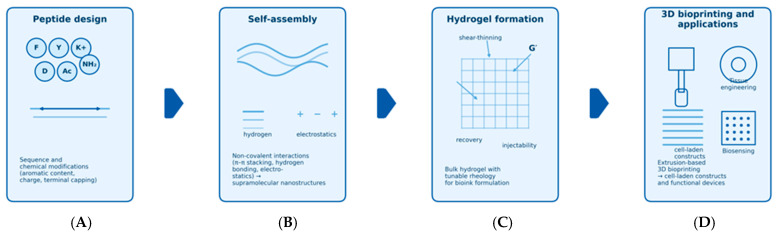
Schematic overview of the USP-based 3D bioprinting workflow. Ultra-short peptide sequences are rationally designed and chemically modified (Panel (**A**)) to promote specific non-covalent interactions and self-assembly into nanostructures (Panel (**B**)). The resulting supramolecular networks form injectable, shear-thinning hydrogels with tunable mechanical properties suitable for use as bioinks (Panel (**C**)). These bioinks can then be extruded to fabricate cell-laden 3D constructs and functional devices for applications in tissue engineering, drug screening, and biosensing (Panel (**D**)). The workflow schematic was drafted with AI assistance (ChatGPT 5.2, OpenAI; accessed 15 December 2025) and finalized and verified by the authors.

**Figure 3 gels-12-00049-f003:**
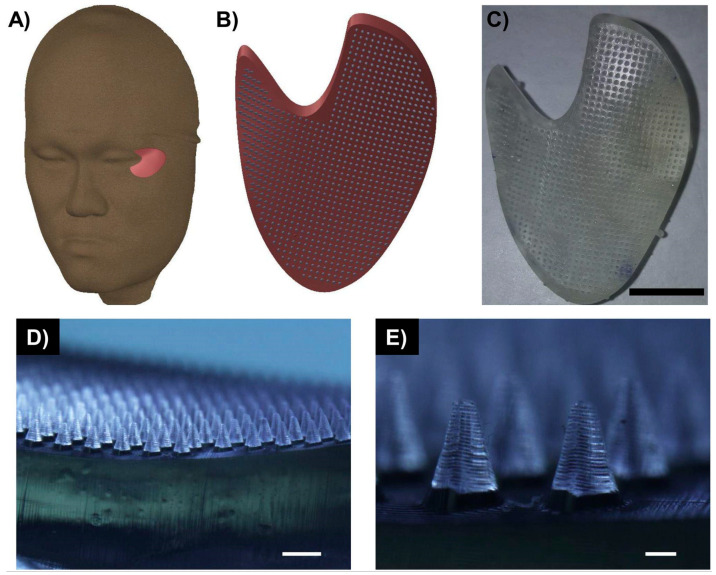
The computer-aided design (CAD) and real photos of a customized 3D-printed MN-based patch. An eye patch based on 3D CAD model outlining the periorbital region in a human volunteer is shown in (**A**), while in (**B**), a CAD model of MN patch is shown. Moreover, (**C**) shows 3D-printed MN with a scale bar of 1 cm, while in (**D**), a microscope image of MN patch with a scale bar of 1 mm is seen. Finally, the MN patch, microscope—based image of 200 μm, is exhibited in (**E**). Reprinted from Lim et al. ©2020 Elsevier B.V. All rights reserved [62].

**Figure 4 gels-12-00049-f004:**
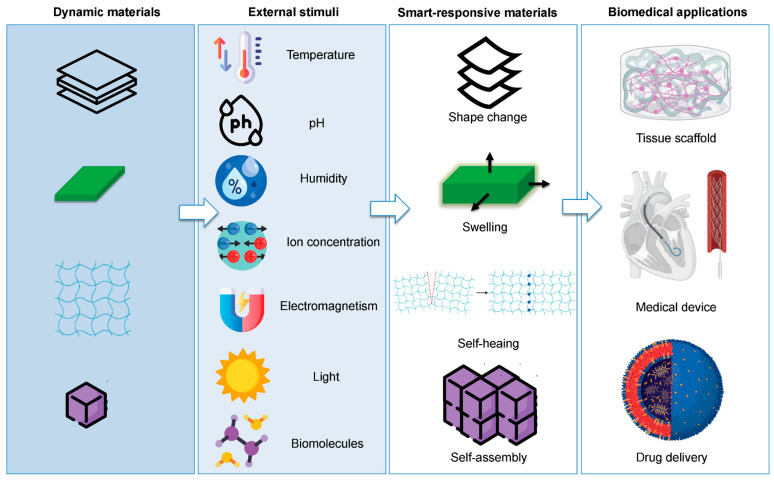
The possible uses of AI in 3D- and 4D printing applications. Reprinted under Open Access Creative Commons license by [72].

**Figure 5 gels-12-00049-f005:**
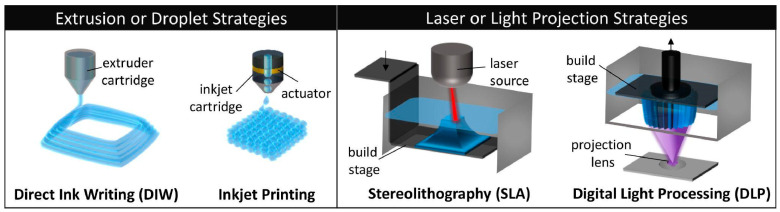
Schematic of conventional 3D printing strategies used for hydrogels. Reprinted under Open Access Creative Commons license by [82].

**Figure 6 gels-12-00049-f006:**
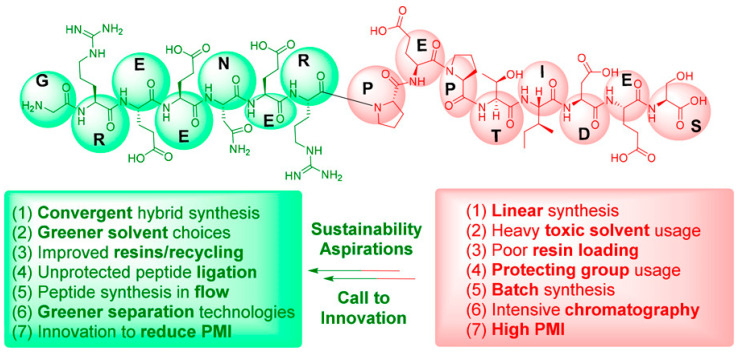
Comparison between conventional and sustainable peptide synthesis strategies. Reprinted under Open Access Creative Commons license by [89].

**Table 1 gels-12-00049-t001:** Representative USP-based bioinks used in 3D bioprinting and their printability and biological performance.

Study/Ref.	Peptide Length	Printing Modality	Key Printability/Mechanical Features	Cell Type(s)	Cell Response
Loo et al. [55]	Hexapeptide USPs	Extrusion-based 3D bioprinting	Instant gelation under physiological conditions; printed scaffolds maintain structural integrity during and after printing	Human mesenchymal stem cells	High viability; differentiation into skin- and intestinal-like organotypic tissue constructs
Arab et al. [56,57]	IVFK, IVZK tetrapeptides	Extrusion-based 3D bioprinting	Good printability: ECM-mimicking nanofibrous networks that entrap aqueous media	C2C12 mouse myoblast cells; muscle myoblast cells	Enhanced adhesion and proliferation; promotion of myogenic differentiation into muscle fibers
Susapto et al. [35]	Tetrapeptides IIFK, IIZK, IZZK	Extrusion-based 3D bioprinting	Transparent hydrogels even at 0.1% *w*/*v*; desirable stiffness and shape fidelity; instant solidification during printing	Human dermal fibroblasts; human bone marrow mesenchymal stem cells	Supports neuron growth; maintains viability and structural integrity over weeks; chondrogenic development
Alhattab et al. [14]	Tetrapeptides IIZK, IZZK	In vivo bioprinting/cartilage fabrication	Self-assembly into nanofibrous hydrogels under physiological conditions; suitable for minimally invasive delivery	Human bone marrow mesenchymal stem cells	Biocompatible scaffolds: differentiation of HBMSCs into chondrocytes
Jian et al. [21]	Dipeptides Fmoc-YD, Fmoc-YK, and mixtures	Extrusion-based printing of hydrogels	Tunable mechanics and biodegradability via concentration and mixing ratio; mixed system reported > 5× strength increase at specific ratios	HepaRG cells (human hepatic cells)	Supports cell growth; tunable degradation and shape fidelity
Rauf et al. [58]	Tetrapeptides IVZK (Ac-Ile-Val-Cha-Lys-NH_2_) and IVFK (Ac-Ile-Val-Phe-Lys-NH_2_)	In situ extrusion-based 3D bioprinting under physiological conditions	Durable constructs; good printability and stability under physiological conditions	Human dermal fibroblasts; human bone marrow mesenchymal stem cells	High biocompatibility; maintenance of viability within printed scaffolds
Khan et al. [43]	Tetrapeptides IVZK/IVFK	Robotic extrusion 3D bioprinting with vacuum assistance	Vacuum-assisted system improves print resolution; enables ~40 mm cylindrical constructs with lower water content and improved stability	Human dermal fibroblasts	Constructs suitable for cartilage tissue engineering
Ahn et al. [59]	PPF-based ink with immobilized RGD tripeptide	Micro-stereolithography	Macroporous 3D scaffolds with controlled architecture; photocross-linkable system provides high shape fidelity	Human chondrocytes	Improved cell–matrix interaction; potential for cartilage tissue regeneration

## Data Availability

No new data were created or analyzed in this study. Data sharing is not applicable to this article.

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
