# Peer review of "Gels2026, 12(1), 49;https://doi.org/10.3390/gels12010049"

_gels, 2026, doi:10.3390/gels12010049_

Round 1

Reviewer 1 Report

Comments and Suggestions for Authors

The overall content and arrangement of this article is okay. It needs a careful edition and revision. For example, the names of the ultra-short peptides (USPs, USP), three-dimensional (3D) and extracellular matrix (ECM) are too chaos both in the Abstract and text. In page 7, line 284, the subtitle “4.1.1 USPs for Tissue Engineering and Scaffolds” lacks logicality. Scaffolds and tissue engineering are not at the same level. Ordinarily, scaffolds are only a small part of the “tissue engineering”. ……

Comments on the Quality of English Language

The quality of English language is normal.

Author Response

Comment: The overall content and arrangement of this article is okay. It needs a careful edition and revision. For example, the names of the ultra-short peptides (USPs, USP), three-dimensional (3D) and extracellular matrix (ECM) are too chaotic both in the Abstract and text. On page 7, line 284, the subtitle “4.1.1 USPs for Tissue Engineering and Scaffolds” lacks logicality. Scaffolds and tissue engineering are not at the same level. Ordinarily, scaffolds are only a small part of “tissue engineering”.

Response: 

We sincerely thank the reviewer for their positive overall assessment of the content and structure, and for these very helpful suggestions. The comments helped us to improve the manuscript in a fruitful way. 

In response, we have undertaken a thorough revision of the manuscript to improve the English language, clarity, and style. In particular, we carefully revised the use of abbreviations throughout the Abstract and main text:

  1. Ultra-short peptides are now introduced once as “ultra-short peptides (USPs)” and consistently referred to as USPs thereafter
  2. Three-dimensional bioprinting is now introduced 3D bioprinting everywhere in the manuscript.
  3. Extracellular matrix is now introduced as “extracellular matrix (ECM)” and consistently referred to as ECM.

We have also taken this opportunity to correct minor typographical and formatting issues, and to streamline several sentences for improved readability (all of them are highlighted). 

Regarding the subtitle on page 7 (line 284 in the original version), we agree that scaffolds should not appear at the same hierarchical level as tissue engineering. We have therefore modified the subtitle from: “4.1.1 USPs for Tissue Engineering and Scaffolds” to “4.1.1 USP-Based Scaffolds for Tissue Engineering” which indeed more accurately reflects that scaffolds are one component within tissue engineering applications.

We believe these revisions have improved the logical structure and overall clarity of the manuscript.

Reviewer 2 Report

Comments and Suggestions for Authors The manuscript offers a relevant, organized, and scientifically based overview of ultra-short peptides (USPs) as novel materials for 3D bioprinting. It connects molecular design concepts, self-assembly properties, hydrogel mechanics, bioprinting methods, and biomedical applications, providing an extensive view of existing research and upcoming possibilities.

Comments on the Quality of English Language The English is adequate for peer review but needs revision by a native English-speaking scientific editor or professional language service prior to final approval.

Author Response

Comment: The manuscript offers a relevant, organized, and scientifically based overview of ultra-short peptides (USPs) as novel materials for 3D bioprinting. It connects molecular design concepts, self-assembly properties, hydrogel mechanics, bioprinting methods, and biomedical applications, providing an extensive view of existing research and upcoming possibilities.

The English is adequate for peer review but needs revision by a native English-speaking scientific editor or professional language service prior to final approval.

Response: 

We thank the reviewer for the very positive assessment of the manuscript’s relevance, organization, and scientific basis. We appreciate the remark regarding the English language.

The manuscript was originally drafted in English by the first author, whose native language is English. Nevertheless, in response to the reviewer’s suggestion, we have carefully re-checked and polished the text to further improve clarity, grammar, and style. During this process, we also standardized the use of abbreviations (e.g., USPs, 3D, ECM) and corrected minor typographical issues throughout the manuscript.

We hope that the quality of the written English in the revised version now fully meets the journal’s standards.

Reviewer 3 Report

Comments and Suggestions for Authors

General Comments The authors have done a good job summarizing the current state of USP hydrogels for 3D bioprinting. The text is readable and the literature coverage is adequate. It’s a good fit for the journal, but the manuscript needs a few specific tweaks to increase its utility for readers.

Specific Points:

  1. Mechanism vs. Description: The text describes what has been done but often misses why it works. Please add more commentary connecting peptide sequence design to the resulting gel mechanics.
  2. Data Presentation: The section on bioprinting is hard to digest in text format. Please summarize the representative bioinks in a table (Sequence vs. Printability/Viability).
  3. Schematic: A figure showing the process (Peptide -> Assembly -> Gel -> Print) would be very helpful for visualization.
  4. Outlook: The "Future Perspectives" section is too light. It needs to realistically address challenges like mechanical robustness and the lack of standardized datasets for AI.

Decision: Accept with Minor Revisions.

Author Response

Comment 1: The authors have done a good job summarizing the current state of USP hydrogels for 3D bioprinting. The text is readable and the literature coverage is adequate. It’s a good fit for the journal, but the manuscript needs a few specific tweaks to increase its utility for readers.

Response 1: We would like to thank the reviewer for the very positive evaluation of our work and for the constructive suggestions. We are pleased that the reviewer considers the manuscript a good fit for Gels and appreciates the readability and literature coverage. We have carefully revised the manuscript in response to all specific points raised, as detailed below. In the revised version, we have implemented all suggested improvements (mechanistic commentary, tabular summary of bioinks, a schematic of the overall process, and a more critical outlook) to further enhance the clarity and usefulness of the manuscript for readers.

Comment 2:  Mechanism vs. Description: The text describes what has been done but often misses why it works. Please add more commentary connecting peptide sequence design to the resulting gel mechanics.

Response 2: We fully agree that making the mechanistic connections more explicit will increase the impact of the review. In the revised manuscript, we have therefore added several passages that directly link peptide sequence and molecular design to self-assembly behavior and gel mechanics. In the paragraphs describing the design principles of ultra-short peptides, we now explicitly discuss how features such as aromatic content, hydrophobic–hydrophilic balance, charge distribution, and N-terminal capping promote specific supramolecular interactions (π–π stacking, hydrogen bonding, electrostatic interactions). We explain how these influence nanofiber density, storage modulus (G′), shear-thinning behavior, and recovery after shear. Moreover, in the paragraph on 3D bioprinting applications, we have revised the discussion of representative USP systems to highlight how small sequence modifications (e.g., substitution of aromatic or bulky residues, introduction of charged residues, co-assembly of complementary sequences) translate into differences in gel stiffness, printability, and shape fidelity of the printed constructs. We have also added a short summarizing paragraph at the end of the bioprinting subsection that explicitly compares these systems and emphasizes how rational sequence design can be used to engineer rheological properties tailored to specific printing modalities and target tissues. We believe now that these additions address the reviewer’s concerns.

Comment 3: Data Presentation: The section on bioprinting is hard to digest in text format. Please summarize the representative bioinks in a table (Sequence vs. Printability/Viability). 

Response 3: We thank the reviewer for this very practical suggestion. To improve readability and facilitate comparison between different systems, we have added a new Table 1 in the section on 3D bioprinting applications. This table summarizes the representative USP-based bioinks discussed in the text and includes, for each system: (i) the peptide (or peptide combination) and sequence/length, (ii) the printing modality (e.g., extrusion-based bioprinting, in situ printing), (iii) key rheological or mechanical features relevant to printability (such as shear-thinning behavior, yield stress, gelation conditions, and shape fidelity), (iv) the main cell type(s) used, and (v) the reported cell response (e.g., cell viability, proliferation, differentiation outcomes).  We believe that this tabular summary makes the bioprinting section much easier for navigation and provides a concise overview of the current USP-based bioinks for quick reference.

Comment 4: Schematic_A figure showing the process (Peptide -> Assembly -> Gel -> Print) would be very helpful for visualization. 

Response 4: We agree that a schematic overview of the entire process will greatly help readers, especially those less familiar with peptide-based materials. In the revised manuscript, we have therefore included a new schematic figure that illustrates the workflow from molecular design to bioprinted construct.

Comment 5. Outlook: The "Future Perspectives" section is too light. It needs to realistically address challenges like mechanical robustness and the lack of standardized datasets for AI.

Response 5: We thank the reviewer for pointing out the need for a more critical and realistic outlook. We have substantially revised and expanded the “Future Perspectives” (and, where appropriate, the Conclusions) to address these points discussing the mechanical robustness and the lack of standardization datasets relevant to AI models. 

After these additions, we believe that we cover the reviewers expectations and we thank him once again for his constructive comments. 

Round 2

Reviewer 1 Report

Comments and Suggestions for Authors

The quality of the manuscript has been improved to a certain degree. However, the format of the abbreviates are still very chaotic. For example, in the Abstract, the second ultra-short peptides (page 1, line16) should be USPs. The 3D bioprinting (page 1, line 17) should be three-dimension (3D) bioprinting, while the three-dimensional networks (page 1 line 18-19) should be 3D bioprinting. ...... A careful revision is still needed. 

Comments on the Quality of English Language

The quality of English language is normal.

Author Response

Comment 1: The quality of the manuscript has been improved to a certain degree. However, the format of the abbreviates are still very chaotic. For example, in the Abstract, the second ultra-short peptides (page 1, line16) should be USPs. The 3D bioprinting (page 1, line 17) should be three-dimension (3D) bioprinting, while the three-dimensional networks (page 1 line 18-19) should be 3D bioprinting. ...... A careful revision is still needed. 

Response 1: We thank the reviewer for the careful reading. We revised the Abstract and manuscript to standardize abbreviations and formatting throughout: “ultra-short peptides” are consistently abbreviated as USPs after first definition, and “three-dimensional (3D)” is introduced at first mention and then used consistently (e.g., 3D bioprinting, 3D networks).

We also corrected minor inconsistencies and typographical issues accordingly.